# Proposal to Test a Transient Deviation from Quantum Mechanics’ Predictions for Bell’s Experiment

**DOI:** 10.3390/e23121589

**Published:** 2021-11-27

**Authors:** Alejandro Andrés Hnilo, Monica Beatriz Agüero, Marcelo Gregorio Kovalsky

**Affiliations:** Centro de Investigaciones en Láseres y Aplicaciones (CEILAP), UNIDEF (MINDEF-CONICET), CITEDEF, J.B. de La Salle 4397, Villa Martelli 1603, Argentina; maguero@citedef.gob.ar (M.B.A.); mkovalsky@citedef.gob.ar (M.G.K.)

**Keywords:** foundations of quantum mechanics, optical tests of quantum theory, entanglement, EPR paradox, Bell’s inequalities

## Abstract

Quantum mechanics predicts correlations between measurements performed in distant regions of a spatially spread entangled state to be higher than allowed by intuitive concepts of Locality and Realism. These high correlations forbid the use of nonlinear operators of evolution (which would be desirable for several reasons), for they may allow faster-than-light signaling. As a way out of this situation, it has been hypothesized that the high quantum correlations develop only after a time longer than *L*/*c* has elapsed (where *L* is the spread of the entangled state and *c* is the velocity of light). In shorter times, correlations compatible with Locality and Realism would be observed instead. A simple hidden variables model following this hypothesis is described. It is based on a modified Wheeler–Feynman theory of radiation. This hypothesis has not been disproved by any of the experiments performed to date. A test achievable with accessible means is proposed and described. It involves a pulsed source of entangled states and stroboscopic record of particle detection during the pulses. Data recorded in similar but incomplete optical experiments are analyzed, and found consistent with the proposed model. However, it is not claimed, in any sense, that the hypothesis has been validated. On the contrary, it is stressed that a complete, specific test is absolutely needed.

## 1. Introduction

Quantum mechanics (QM) has faced debate since its early years. At least three points of conflict have been identified:(i)Failure of the principle of correspondence in chaotic systems (“quantum chaos”).(ii)Evolution from a superposition state to the observed state of a system is not described by the theory (the “measurement” or “projection” problem).(iii)Correlations between measurements performed in distant regions of a spatially extended entangled state are higher than allowed by Local Realism (violation of Bell’s inequalities).

Local Realism (LR) is shorthand for the intuitive notions of separability of physical phenomena, and that the features of these phenomena are independent of being observed or not [1,2]. LR is assumed not only in everyday life, but also in all scientific practice (except QM).

The conflictive point (ii) can be solved by using nonlinear operators of evolution [3,4]. However, it has been shown [5,6] that such operators would allow faster-than-light signaling, if combined with the high correlations of point (iii). This would be in contradiction with theory of Relativity, and make quantum field theory untenable. The conclusion is that any nonlinear extension of QM is impossible, even if it is infinitesimal. This conclusion has been interpreted in two opposite ways. For some, it is the demonstration that QM is part of the “ultimate theory” of Nature, for the introduction of nonlinear correction terms is impossible. For others, instead, it means that the predictions of QM are structurally unstable, and hence that there is something important missing in its formulation.

Hope to reconcile QM with LR comes up by noting that the performed tests leave space for the following hypothesis: QM predictions are valid for statistical averages measured over “long” times; measurements performed in “short” times would hold to LR. The time scale here would be given by the time light needs to cover the spatial spread *L* of the entangled state. In other words, QM (as we know it) would describe t→∞ stationary states, while transient deviations from QM predictions would be observed in times shorter than *L*/*c*. If this hypothesis was demonstrated correct, it would solve not only point (iii), but it would also open the door to the use of nonlinear operators to solve point (ii) (it would probably allow a more appropriate approach to point (i), too). Faster-than-light signaling would never be possible, for the high correlations characteristic of entanglement would exist only after a time longer than *L*/*c* had elapsed. Note that usual QM would still make correct predictions in the overwhelming number of cases. The only case where we foresee some limitation is Quantum Key Distribution using entangled states. In principle, the transient deviations would impose the rate of detected entangled pairs to be smaller than *c*/*L*. Anyway, this limit is several orders of magnitude above the rate that can be reached nowadays (see Section 3).

Following this hypothesis, in Section 2, we introduce a simple hidden variables model named “AB”. Similar to almost all such models, it is an artificial construction. It is not claimed to provide an accurate description of an underlying physical reality. It has a well-defined form in order to be testable, but we expect it to reveal the general features of the hypothesized transient deviations. In Section 3, an achievable testing experiment is explained. It is based on stroboscopic reconstruction of time variation of entanglement and efficiency. In Section 4, we analyze data obtained in similar but incomplete experiments performed between 2012 and 2019. These data are consistent with AB, but we do not claim them to provide any sort of confirmation of the transient deviation hypothesis—they just encourage performing the appropriate experiment.

## 2. The AB Model

### 2.1. A Hidden Variables Theory

Because of its relevance and simplicity, the case of two photons entangled in polarization is considered (see Figure 1). Source S emits pairs of photons in the fully symmetrical Bell state: ∣φ^+^〉 = (1/√2) × {∣x_A_,x_B_〉 + ∣y_A_,y_B_〉} towards stations A and B separated by a distance *L*. Entanglement is evaluated, f.ex., with the Clauser–Horne–Shimony and Holt parameter S_CHSH_ [1]. It involves measuring the probability (defined as the frequency limit) of coincident detections at each pair of gates at each station: P^++^(*a*,*b*), P^−−^(*a*,*b*), P^+−^(*a*,*b*) and P^−+^(*a*,*b*), with angle settings *a* = {0,π/4} and *b* = {π/8,3π/8}. The QM ideal prediction is S_CHSH_ = 2√2, while LR imposes S_CHSH_ ≤ 2.

An important parameter is *efficiency* η ≡ N_c_/N_s_, where N_c_ is the number of coincidences and N_s_ is the number of single detections in a given run. It is well known that hidden variables holding to LR can exactly reproduce QM predictions if η ≤ 1/√2 ≈ 0.71 [7].

We assume that each emitted pair carries an angular “hidden variable” α that takes values in [0,2π] and has two equally probable sub-states, α^+^ and α^−^. P*_i_*^+^ (P*_i_*^−^) is the probability to detect a photon at gate “+” (“−”) in station *i*. The probabilities of detection at station A (setting *a*) are assumed to be:if α = *a* and α = α^+^, P_A_^+^ = 1 and P_A_^−^ = 0;
if α = *a* and α = α^−^, P_A_^+^ = 0 and P_A_^−^ = 1;
if α ≠ *a*, P_A_^+^ and P_A_^−^ = 0.(1)
and at station B:if α = α^+^, P_B_^+^ = *cos*^2^(*b −* α) and P_B_^−^ = *sin*^2^(*b −* α)
if α = α^−^, P_B_^+^ = *sin*^2^(*b −* α) and P_B_^−^ = *cos*^2^(*b −* α)(2)

(regardless the value of *a*, which is unknown at B).

The probability of coincidence in the “+” gates is then:if α = α^−^, P_B_^+^ = *sin*^2^(*b −* α) and P_B_^−^ = *cos*^2^(*b −* α)(3)
where *p*(α = *a*) is the probability that α = *a* (the factor ½ comes from the two equally probable sub-states of α). This result is proportional to the QM prediction. The same holds for the coincidence probabilities P^+−^, P^−+^, and P^−−^, so that S_CHSH_ = 2√2 is obtained. However, there is an asymmetry in the number of single counts in each station: “all” emitted photons are detected at B, while only “some” of them are detected at A. To be precise: if R^α^ is the total rate of detections (regardless if + or −) at B, the rate at A is only *p*(α = *a*) × R^α^. In order to balance the rates of singles at each station, we assume that half of the emitted pairs carry a hidden variable β that mirrors Equations (1) and (2):if β = *b* and β = β^+^, P_B_^+^ = 1 and P_B_^−^ = 0;
if β = *b* and β = β^−^, P_B_^+^ = 0 and P_B_^−^ = 1;
if β ≠ *b*, P_B_^+^, P_B_^−^ = 0.(4)
and:if β = β^+^, P_A_^+^ = *cos*^2^(*a −* β) and P_A_^−^ = *sin*^2^(*a −* β)
if β = β^−^, P_A_^+^ = *sin*^2^(*a −* β) and P_A_^−^ = *cos*^2^(*a −* β)(5)

Choosing *p*(α = *a*) = *p*(β = *b*) ≡ *p*, the same rate of singles is detected at each station. The probability of detecting a single photon at A is: *probability of being an* α*-pair* × *p*(α = *a*) + *probability of being a* β*-pair* × 1 (for, all β-pairs are detected at A) = ½*p* + ½. In turn, the probability of detecting a coincidence is ½ × *p*(α = *a*) + ½ × *p*(β = *b*) = *p*. Therefore, the efficiency is:η = 2*p*/(1 + *p*) (6)

In the case of experiments with fast and random variation of analyzers’ settings [8,9], there are two alternatives in each station, so that *p* = ½ ⇒ η = ⅔ (and S_CHSH_ = 2√2). In the general case, it is convenient to relax the strict equalities in Equations (1), (2), (4) and (5) to:P_A_^+^ or P_A_^−^ = 1 if α ∈ [*a* − Δ, *a* + Δ]
P_A_^+^ or P_A_^−^ = 1 if α ∈ [*a* − Δ, *a* + Δ]
P_A_^+^ and P_A_^−^ = 0 otherwise.(7)

P_B_^+^ and P_B_^−^ are defined as in Equation (2) (and symmetrically for β).

Now, *p* = Δ/π. If Δ = π, then *p* = 1 and η = 1, but P^++^ = ¼ (no correlation). After integrating α and β in [0,2π], the total probability of “++” coincidence is:P^++^ = (1/4π){*sin*(2Δ) × *cos*^2^(*a − b*) + Δ – ½ *sin*(2Δ) }(8)

If Δ ≪ 1, then P^++^ ≈ (Δ/2π) × *cos*^2^(*a − b*) = *p* × ½ × *cos*^2^(*a − b*), as in Equation (3). The expressions for the coincidence probabilities P^+−^, P^−+^, and P^−−^ are analogous to Equation (8). It is found that:S_CHSH_ = 2√2 *sin*(*x*)/*x*(9)
where *x* = 2Δ. Equations (8) and (9) are valid even if there is no correlation at all between the values of α and *a* (β and *b*). That is, they are valid even for the experiments performed with fast and random variation of the settings α and β.

### 2.2. Delayed Reaction

At this point, the model simply exploits the efficiency loophole [1,7]. We now add that the detection (in general: absorption) of a photon at station A produces a “reaction” on the hidden variable. This reaction propagates backwards to the source, carrying information on properties the photon had when it was detected (in the case of interest here, whether it was detected polarized with an angle parallel, or perpendicular, to *a*). The reaction arrives to the source at time τ ≥ *L*/*c* after the emission (τ/2 to go and τ/2 to return), and influences the values of α emitted thereafter, in such a way that they tend to fit α = *a*; that is, a sort of stimulated emission at the hidden variables level. The reaction is proportional to the rate of detected photons R_det_ and to the mismatch between α and *a* when the photon was detected (that is, when the reaction was “born”). Therefore, the evolution of the values of α emitted at time t is given by:dα(t)/dt ∝ −R_det_(t − τ/2)×[α(t − τ) − *a*(t − τ/2)](10)

R_det_(t − τ/2) ∝ R_emitted_(t − τ), so that:dα(t)/dt = −Γ(t − τ)×[α(t − τ) − *a*(t − τ/2)](11)
where the value of Γ is unknown, but is supposedly proportional to the rate of emitted photons (which is, in turn, proportional to the source intensity). Before the delayed reaction arrives, the source is assumed to emit random values of α. If it emits entangled states in the Bell state ∣φ^+^〉, the beams propagating to A and B carry the same value of α. If the source is prepared to emit classical states instead, the values of α in each beam are not correlated. An expression similar to Equation (11) holds for β(t).

The hypothesized reaction is inspired by the absorber theory of radiation proposed by Wheeler and Feynman [10]. In this theory, the reaction propagates backwards not only in space but also in time. D. Pegg shows this theory to explain the experiment in Figure 1 in an elegant way [11]. The theory is impeccable, but the idea of signals propagating backwards in time is uncomfortable. The delayed reaction hypothesized here has a poorer theoretical basis, but is free of that uncomfortable feature. It clearly holds to Locality and, most importantly, leads to a testable difference with QM.

Equation (11) is a delay differential equation. It evolves in a phase space of infinite dimensions. Its solutions are difficult to find in the general case. In the particular case that Γ(t) and *a*(t) are constant, the solutions have the general form α(t) = α_0_.*exp*(zt/τ), z ∈ **C**. Critical damping of α(t) towards *a* is reached for Γτ = 1/*e*. For 1/*e* < Γτ < π/2, α(t) converges towards *a* with damped oscillations of period ≈4τ. S_CHSH_ is given by Equation (9) for all times, while η varies on time depending on the distance between α(t) and *a*. When α(t) ∈ [*a* − Δ, *a* + Δ], η = 1 (Equation (7)). If t→∞, then α→*a*, β→*b*, *p*→1, η→1, S_CHSH_→2√2, and *all* observables coincide with QM predictions regardless of the value of Δ. For Γτ ≥ π/2, the oscillation amplitude of α(t) diverges. In this case, α(t) runs away, losing any correlation with *a*. Equation (9) for S_CHSH_ still holds, and η = ⅔ as in the case of setups with random and fast variation of the settings.

Experiments focused on measuring time-averaged magnitudes cannot provide a clear-cut disproval of the transient deviation hypothesis, even if they are essentially loophole-free [12,13,14,15]. These experiments reach η > ⅔, but this result can be reproduced by a modified version of AB exploiting remaining imperfections. Such a modified version is more involved and uninteresting, for it is always possible to design a classical theory able to fit the numbers of a specific experimental realization [16]. This does not lessen the significance of the loophole-free experiments. They did reach the goals they were designed for [17]. The point is that no real experiment is perfect, and that experimental results cannot be reliably applied to test a model different from the one they were designed to test. The transient deviation hypothesis can be tested in an easy and definitive way by observing time evolution of efficiency and/or entanglement.

To explore the behavior of Equation (11) further, we can run numerical simulations. In Figure 2, Γ is “turned on” at t = 0 and remains constant thereafter; we choose *a* = π/4 and α(t = τ) = 0. Values of α(t) for t < τ are random in [0,2π]; they are not plotted to not to burden the figure. At t = τ, the reaction starts to influence the emitted values of α. Yet, as the values α(t – τ) in the rhs of Equation (11) are randomly distributed, α(t) does not evolve far from α(τ) during this stage. At t = 2τ = 1000, the values of α(t − τ) are not random any longer, and α(t) starts to evolve towards the “target” value *a*. It does so in a monotonous or oscillatory way, depending on the value of Γ.

An interesting numerical result is that (for the same value of Γ) oscillations cross their target values (0 and π/4) at nearly the same time even if the initial values α(t ≤ τ) are all different (Figure 3). The only condition for this to occur is that the random initial values are distributed symmetrically to the target. As they are assumed uniformly distributed in [0,2π], the condition is fulfilled for any target value. As shown later, this effect of *spontaneous synchronization* has important consequences.

The hidden variable α(t) cannot be directly observed. The value of S_CHSH_ is fixed by Equation (9) and depends on Δ weakly. Perhaps surprisingly, the only observable effect comes up in the time variation of η. In the next section, an experiment to test the transient deviation hypothesis (of which the AB model is probably the simplest expression) is described.

## 3. A Proposal to Test the Transient Deviation Hypothesis

### 3.1. Difficulty of Direct Observation

According to the usual description of Figure 1 experiment, there is no reason to expect any time variation of efficiency (in ideal conditions, of course). A straightforward approach to test AB is hence to measure η(t) during times shorter than τ. Yet, such measurement is impossible nowadays. In order to check entanglement and observe η(t) with enough detail, a rate of coincidences >10^3^ τ^−1^ is required. The highest reported rate is ≈3 × 10^5^ s^−1^ in a laboratory environment [18], which means ≈10^−4^ τ^−1^. Numbers are better for larger *L*: 50 s^−1^ at 13 Km [19] (≈2 × 10^−3^ τ^−1^) and 8 s^−1^ at 144 Km [20] (≈4 × 10^−3^ τ^−1^), but still short by almost six orders of magnitude.

One possibility at hand is a “stroboscopic” approach. It is reasonable to expect α(t) to decay to a random distribution in a finite time τ_d_ after the driving reaction is turned off. A stroboscopic measurement of η(t) is then possible, by using a pulsed source of pairs with time between pulses longer than τ_d_. Each pulse produces at most one photon recorded (at random time position within the pulse), and the behavior of η(t) and S_CHSH_(t) is reconstructed after millions of pulses. As in all stroboscopic reconstructions, it is necessary to assume that the system decays to the same “ground state” after each pulse. However, in this case, that state means random α(t), so it is not a stringent assumption. Although τ_d_ is unknown, pulse repetition rate R_p_ can be lowered until some effect is perceived.

An observation suggests that τ_d_ might actually be quite short: a performed pulsed version of the Figure 1 setup (see details in Section 4) shows that if the time coincidence window defining coincidences is increased beyond the pulse duration, the measured total S_CHSH_ parameter decays following the curve calculated assuming that detections outside the pulse are fully uncorrelated [21]. Assuming non-correlation implies the curve, but, of course, observing the curve does not necessarily imply actual non-correlation. Nevertheless, if the last implication is accepted valid, then τ_d_ would be shorter than the time resolution of that experiment, i.e., τ_d_ < 12.5 ns, and hence, values of R_p_ as high as 80 MHz would be acceptable. This figure is far higher than estimated necessary to perform the experiment (see next Section).

### 3.2. Description of the Proposed Setup

We detail the proposed experiment:(i)*The source of pairs*: a set of nonlinear crystals is pumped with square-shaped laser pulses of rise time and fall time τ_rf_ and full duration τ_pulse_, equally separated by a time R_p_^−1^. All these parameters, and also the pump intensity, are adjustable. A laser diode at 405 nm pumping a pair of two crossed type-I phase-matched nonlinear crystals [22] seems to be the simplest choice.(ii)*The stations A and B* are provided with devices to record the time of detection of each photon (time stamping), and also the time of emission of each pump pulse (to determine the starting time of the stroboscopic reconstruction) with resolution τ_res_. Samples of the pump are sent to each station to synchronize the starting time of each pulse. In some experiments devoted to test QM vs. LR, the analyzers’ settings {*a,b*} are varied randomly in a time shorter than τ to enforce the lack of correlation between the hidden variables and the settings. The assumed decay of α(t) and β(t) to a random state implies that the lack of correlation occurs spontaneously. This not only means an important experimental simplification, but it also solves the problem of ensuring settings’ randomness, which is a sort of infinite regress [23,24,25].Measurement runs are repeated with different values of *L* to scan the time scale.(iii)*Time hierarchy*: the following relationships should hold:τ_res_ ≪ τ_,_ to resolve details of η(t).τ_rf_ ≪ τ, to allow pump pulse to be square shaped.τ ≪ τ_pulse_, to give enough time to the evolution of α(t).τ_pulse_ ≪ R*_p_*^−1^, to allow pulses to be well separated.τ_d_ ≪ R*_p_*^−1^, to allow α(t) to decay to the regime of random emission.

This time hierarchy is summarized in Table 1.

The first critical time in this hierarchy is τ_res_. Jitter in avalanche photodiodes (the usual devices for single photon detection) limits τ_res_ to ≈2 ns. Standard laser diodes and driving devices also reach τ_rf_ ≈ 2 ns. These numbers imply values for τ in the 20–200 ns range (meaning *L* from 6 to 60 m), and hence, τ_pulse_ from 0.2 to 2 μs, and R_p_ from 500 to 50 KHz. All these figures are easily attainable.

Pump intensity must be kept low, so that probability of detection of one photon per pulse remains small. This is to limit the number of spurious coincidences, which is an issue in the pulsed regime [26]. As a consequence, a single-photon detection rate would be expected in the 5–50 KHz range.

### 3.3. Predicted Observations

Numerical simulations of AB are run assuming that Γ is turned on at t = 0 and remains constant thereafter. The sharp “slit” assumed in Equation (7) is simple but rather unphysical; in these simulations, the following smooth condition is used instead:*p*(α = *a*) = *exp* –[α(t)−*a*]^2^/Δ^2^(12)

(and the same for β, *b*). From Equation (6), the efficiency at station A is now:η_A_(t) = {*exp* − (α − *a*)^2^/Δ^2^ + *exp* − (β − *b*)^2^/Δ^2^}/{1 *+ exp* − (α − *a*)^2^/Δ^2^}(13)

(and a symmetrical expression for η_B_), where α and β are functions of time ruled by Equation (11).

The program is iterated many times and results for each time value are summed up, imitating the stroboscopic reconstruction. A numerical iteration of the program corresponds here to a pulse in the actual experiment. Each iteration starts from a different set of random initial conditions for α(t) and β(t), hence, there is no memory from one iteration to the next (what corresponds to pulse separation longer than τ_d_). After many iterations, a curve of η(t) averaged over a large number of different sets of initial conditions, <η>(t), is obtained. The right number of iterations is found by increasing it until the curve of <η>(t) stabilizes. In all tested cases, 10^3^ iterations suffice.

Some results are shown in Figure 4. In spite of the explicit functional dependence in Equation (13), numerically obtained values of <η>(t = τ) are independent of {*a*,*b*}. This is a consequence of the random and symmetrical distribution of α(t) and β(t) for t < τ (which is also the cause of the spontaneous synchronization effect mentioned before). For Δ fixed, the curves start with the same value of <η>(τ) regardless of the value of Γ. This is because the reaction, no matter how weak or strong, has had no time to propagate from the stations to the source. Naturally, <η>(τ) is smaller for smaller Δ (which means a narrower “slit”). As in Figure 2, curves remain near <η>(τ) for a time τ (i.e., until the earliest reaction arrives to the source). Then, α(t) and β(t) start to evolve towards their target values *a* and *b*. The curves differ depending on the value of Γ. The evolution is faster, and the saturation value (=1 in these ideal conditions) is reached sooner, for higher Γ.

If the value of Γ is such as to produce oscillations of the hidden variables, a peak of efficiency is reached each time α(t) and β(t) are close to *a*,*b*. These peaks are clearly visible in Figure 4 in spite of the different initial conditions and target values, thanks to the spontaneous synchronization effect. For small Γ, instead, a nearly linear increase in <η>(t) is visible. If Δ is relatively large, the efficiency saturates sooner and the concavity of the curve is downwards. If Δ is small, it is upwards instead. For large Γ, oscillations are sharper and show higher contrast as Δ decreases, but the period remains the same. This period is proportional to *L*.

The curves in Figure 4 assume the source is equidistant from each station. Dr. Siddarth Joshi put forward the interesting question of what is observed if the stations are placed at different distances. Now, two time scales are relevant: τ_A_ = *L*_A_/*c* and τ_B_ = *L*_B_/*c*, where *L*_A_ (*L*_B_) is the distance from S to the A (B) station. There are also two possible interaction strengths: Γ_A_ and Γ_B_. In the symmetrical case, Γ is scaled with τ, and hence there is a single-parameter set of solutions (leaving aside Δ). In the non-symmetrical case instead, the other interaction strength and time delay have independent values, and hence there is a three-parameter set of solutions. An exhaustive exploration of this 3-D parameter space is beyond the scope of this paper. A couple of relevant cases are discussed in the Appendix A.

If the transient deviation hypothesis is correct, the following phenomena should be then observed:(I)*At fixed L*: <η>(t) increases with time until it saturates. Time needed to reach the saturation value decreases if source intensity is increased. At high intensity, oscillations may appear. If R_p_ is increased above a certain threshold (the unknown value τ_d_^−1^), <η>(t = 0) would also start to increase, for α(t) and β(t) would have had no time to decay between pulses. In the limit case of a continuous source, <η> = 1.(II)*At variable L*: the time <η>(t) takes to reach saturation, and also the period of the oscillations (if they exist), increase with *L*.

These features warrant a test of QM vs. LR different from all the ones attempted until now. It is independent of the violation of a statistical correlation limit. It is also more robust, for a definite result is obtained even if the experimental setup is far from ideal regarding efficiency of optics, detectors, and alignment. The key is the scaling of the evolution of <η>(t) with *L*.

## 4. Results Obtained in Incomplete Experiments

In the year 2012, our group tested and closed the *time coincidence loophole* [21,27]. That experiment can also be regarded as an incomplete version of the one proposed in Section 3. It involved a pulsed source of entangled photons and time-stamped record of both photon detection and pulse emission events, as required, but it did not fit the time hierarchy described in (iii). In particular, τ_res_ ≫ τ, and pump pulses were not square shaped. Most importantly, the values of pump intensity, τ_pulse_, *L,* and R_p_ were not varied.

In Figure 5, stroboscopically measured <η>(t) and S_CHSH_(t) are shown. Entanglement is constant during the whole pulse duration, in agreement with predictions of both QM and AB. Efficiency, instead, increases almost linearly, starting with ≈3% and reaching ≈16% some 100 ns (or ≈400 τ) later. This variation is not predicted by the usual description and is therefore caused by some experimental artifact or by an effect of the kind expected according to the transient deviation hypothesis.

The curve of <η>(t) in Figure 5 sums up coincidences recorded for all {*a*,*b*} settings in order to improve the statistics. The same variation (i.e., nearly linear, and with the same slope) is observed in each subset of data corresponding to each setting {*a*,*b*}. The values in Figure 5 are consistent with Δ < 0.33 (from Equation (9)) and Γ > 1.38 × 10^−3^ ns^−1^. The experiment was repeated in 2018 using the same setup and the nearly linear variation of <η>(t) was observed again, although with a slightly different slope.

In 2019, the experiment was repeated with an improved setup [28]. Distance between stations was still short (τ = 0.5 ns), but time resolution (τ_res_ = 2 ns), pulse duration (τ_pulse_ = 500 ns), and shape (square, τ_rf_ ≈ 2 ns) were closer to the proposal. Single photons were detected at a rate ≈ 60% higher than in 2012, and their wavelength was 810 nm instead of 710 nm.

Now, <η>(t) is observed as constant during the whole pulse duration (also entanglement, S_CHSH_ ≈ 2.67) (see Figure 6). We believe the linear variation in Figure 5 to be a numerical effect (a sort of aliasing) caused by the marginal value of the time resolution compared with pulse duration and the typical time of pulse variation.

Nevertheless, the new observations do not rule out the transient deviation hypothesis completely, for τ is still too short. They are consistent with the AB model with Δ < 0.29 and Γ > 0.08 ns^−1^ (from *d*<η>/*dt* > 0.062 ns^−1^ estimated from Figure 6, left).

In all these experiments, τ_res_ > τ. Oscillations, if they existed, would have not been observed. This limitation leaves space for additional sets of values {Δ,Γ} to be consistent with the observations (yet, these additional values can be found only through numerical trial and error). An experiment where τ_res_ ≪ τ is absolutely needed to test the transient deviation hypothesis.

## 5. Conclusions

The AB model is the simplest realization we found of the general hypothesis of the existence of transient deviations from QM predictions. It is a classical model based on the assumed existence of a reaction on the hidden variable α(t) each time a photon is absorbed. That reaction propagates back to the source in normal time (as opposed to the Wheeler–Feynman theory, where the reaction propagates backwards in time) and influences the future values taken by α(t) in such a way that α(t) converges to the value *a* set for the polarization analyzer. The model is able to reproduce all QM predictions after a time much longer than *L*/*c* has elapsed since the moment the source of entangled states is turned on. The transient deviation hypothesis has not been reliably refuted by experiments performed until now, but it can be clearly tested in a relatively simple setup. The magnitude to measure is stroboscopic time variation of efficiency, and how this variation scales with the distance between stations. Data recorded in similar but incomplete experiments performed in 2012, repeated in 2018, and improved in 2019 are consistent with the transient deviation hypothesis, but they are insufficient to validate or refute it.

The observations described in (I) and (II) in Section 3 define a test of QM vs. LR of a new type, independent of the violation of an average correlation limit. In the usual tests (i.e., the violation of a Bell’s inequality), threshold values of detectors’ efficiencies, angle settings’ unpredictability, and event-ready signals must be ensured to close all the known loopholes. In the test proposed here, instead, a definite answer is obtained even if those difficult technical requisites are not fulfilled. In the usual tests, a low correlation, e.g., S_CHSH_ < 2, cannot be interpreted as a refutation of QM, for it may be the consequence of a poor realization. In the test proposed here, instead, the observation of (I) and (II) would provide support to the transient deviation hypothesis even in an imperfect setup. The usual tests of QM vs. LR face the problem of the never-ending proliferation of loopholes (and hence, technical conditions are increasingly difficult to achieve), never reaching fully conclusive results. The experiment proposed here provides a definite answer from both possible outcomes: if *L*-dependent <η>(t) dynamics are observed, QM (as we know it, at least) will be demonstrated as incomplete. The door will be open to potentially fruitful nonlinear extensions of QM. If such *L*-dependent dynamics are not observed instead, the transient deviation hypothesis will be refuted.

Finally, the proposed experiment is technically simple: the analyzers can be kept fixed, and efficiency and correlation are not needed to reach a high threshold. We believe it difficult to find an experiment at hand with more important potential consequences.

## Figures and Tables

**Figure 1 entropy-23-01589-f001:**
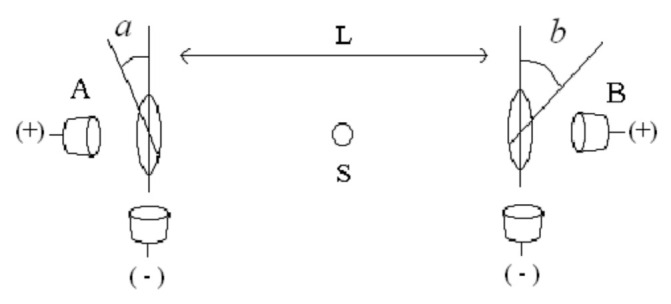
Source S emits entangled states ∣φ^+^〉 towards stations separated by a distance *L*. Analyzers are set at angles *a* and *b*, and single photons are detected transmitted (+) or reflected (−) at each station.

**Figure 2 entropy-23-01589-f002:**
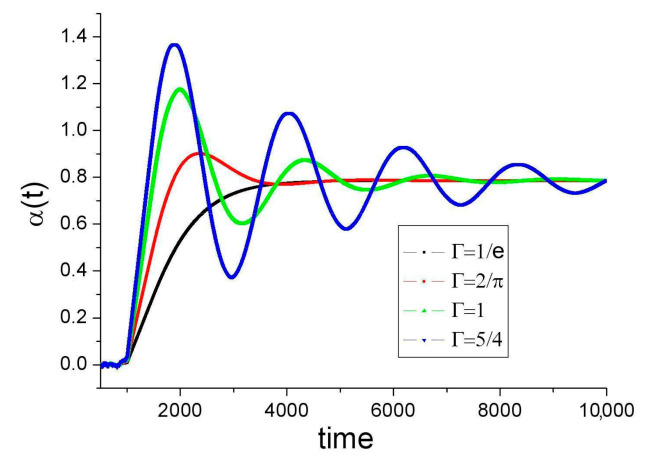
α(t) for several values of Γ scaled with τ (note that the plot starts at t = τ) according to Equation (11); Γ(t < 0) = 0, Γ(t ≥ 0) = Γ, α(t < τ) = random (not plotted), α(t = τ) = 0, *a* = π/4, time scale: τ = 500. For critical damping (Γ = 1/*e*), the value 0.9 × *a* is reached in ≈3.5 τ. Recall that α(t) is a hidden variable; this curve cannot be directly observed.

**Figure 3 entropy-23-01589-f003:**
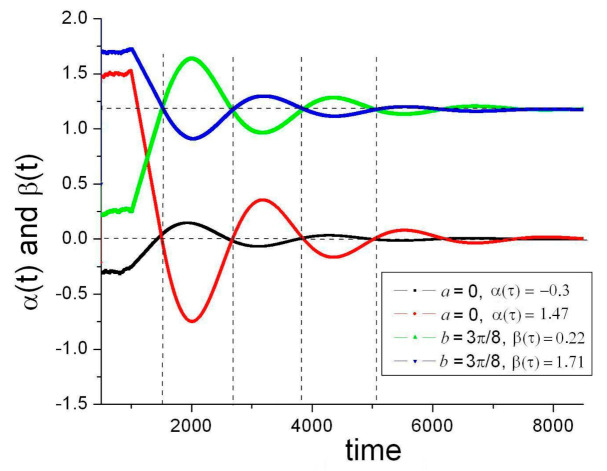
An example of spontaneous synchronization. The evolution of α(t) and β(t) are plotted for different initial conditions (including 500 different random values for t < τ, not plotted) and target values (*a* = 0, *b* = 3π/8), Γ = 1. In spite of all these differences, target values are crossed at nearly the same times. Synchronization improves as the partition of τ increases (here, τ = 500).

**Figure 4 entropy-23-01589-f004:**
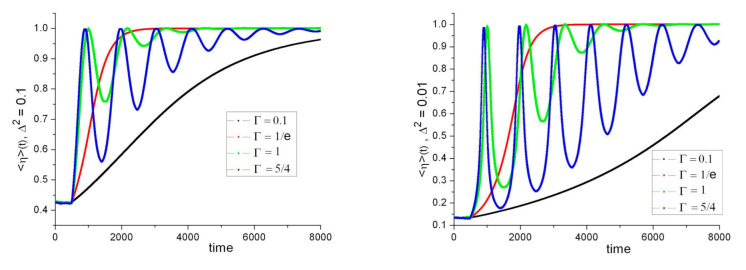
Observable <η>(t) averaged over 10^3^ iterations with different random initial conditions and values of Γ (scaled with τ), Δ^2^ = 0.1 (**left**), Δ^2^ = 0.01 (**right**), time scale: τ = 500.

**Figure 5 entropy-23-01589-f005:**
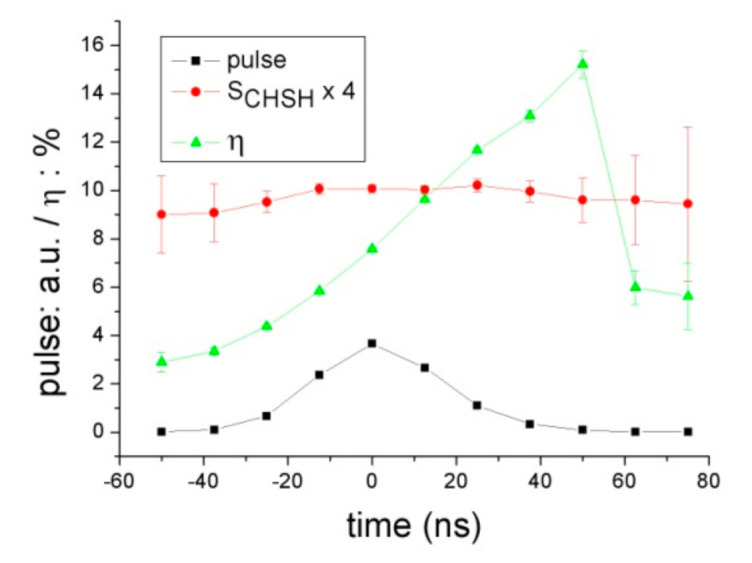
Observed <η>(t) (in %) and S_CHSH_(t) (×4, for clarity of figure) in the 2012 experiment [21], τ_res_ = 12.5 ns, τ = 0.27 ns, R_p_ = 60 KHz, station B. Single counts are also plotted (in arbitrary units), to indicate pulse duration (35 ns FWHM), shape, and position. The best fit to the (assumed linear) slope of <η>(t) is 1.2 × 10^−3^ ns^−1^, average S_CHSH_ = 2.635.

**Figure 6 entropy-23-01589-f006:**
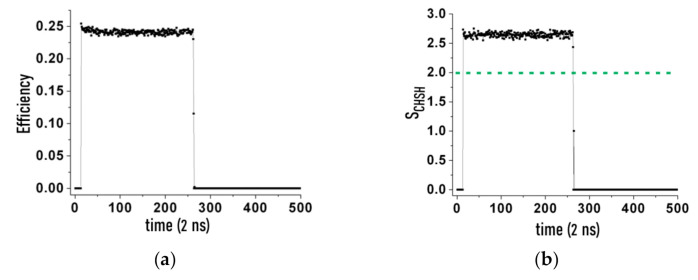
Measured <η>(t) (**a**) and S_CHSH_(t) (**b**) in the 2019 experiment [28], τ_res_ = 2 ns, τ_pulse_ = 500 ns, τ = 0.5 ns, R_p_ = 50 KHz, station A. Horizontal dotted line indicates the limit given by CHSH inequality.

**Table 1 entropy-23-01589-t001:** Summary of the time parameters involved in the proposed experiment. All are adjustable by the observer, except τ_d_, whose value is unknown. Typical or estimated values are indicated between parentheses (see text).

Time Parameter (Name)	Description
τ (20–200 ns)	Time between stations, *L*/*c*
τ_res_ (2 ns)	Resolution of time stamping
τ_pulse_ (200 ns–2 μs)	Pulse duration
τ_rf_ (≪τ_pulse_)	Pulse rise and fall time
τ_d_ (<12.5 ns?)	Decay time to initial state
R_p_^−1^ (>τ_d_)	Separation between pulses

## Data Availability

Measured data are not publicly available at this time.

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
