# Peer review of "Proposal to Test a Transient Deviation from Quantum Mechanics’ Predictions for Bell’s Experiment"

_entropy, 2021, doi:10.3390/e23121589_

Round 1
Reviewer 1 Report
Even, the title of the paper has grammatical error and not understandable. What authors want to say is highly confusing. Neither they properly explain the local realism nor quantum theory. As an example, authors stated in the Abstract (and in many places in the paper), " These high correlations (quantum violation of local realism) forbid the use of nonlinear operators of evolution, which would be desirable for several reasons, for they may allow faster-than-light signaling". Parenthesis is mine. What is the meaning of this kind of sentence. There are many such crazy statements throughout the paper and all of them are wrong. I never heard low velocity and high velocity concept in he context of Bell's theorem. Perhaps I'm not experienced enough. However, as far as my knowledge goes, this work should never be published in any journal. I would suggest the author to first read the meaning of Bell's theorem and implication in a proper way. There will be many potential problems they may be able to identify to criticize Bell's theorem but this is not at all the way. I would also request the Editor to not to entertain such kind of papers in future.Author Response
Please see the attachement.

Reviewer 2 Report
The authors propose a hidden-variable model which, according to them, would be able to reconcile local realism with quantum mechanics. According to the authors, the proposed model has not been refuted by the published experiments that addressed the Bell test loopholes.
The central element of the proposed model is a reaction of a detection event on the source of photon pairs. This reaction influences the values of the hidden variables alpha emitted thereafter in such a way that the hidden variables alpha tend to fit the measurement setting a (page 3). The explanation from the authors for the observed violation of Bell’s inequality makes use of both the efficiency loophole and a version of the memory loophole.
However, the proposed hidden-variable model fails to account for the fact that in several of the published Bell tests the measurement settings were randomly varying and therefore unpredictable. Settings were chosen in such a time interval that no information from the measurement stations (about the measurement settings) was able to reach the source before the photons were emitted.
Therefore, my question to the authors is: How can the source be influenced to produce hidden variables that fit the measurement settings if the measurement settings are randomly varying, unpredictable, and therefore completely unknown to the source? A way out seems to be a form of superluminal signaling, but exactly that is forbidden in local realism.
So how can hidden variables at the source be adjusted to the measurement settings in a way that is compatible with local realism? In my opinion, without an answer to that question, the conclusions of the manuscript are not valid.
Author Response
Please see the attachement

Reviewer 3 Report
The authors propose to test a hypothesis of the existence of transient deviations from QM predictions by use of a classical model based on the assumed existence of a reaction each time a photon is absorbed.
The model reproduce all QM predictions after a time much longer than L/c
has elapsed since the moment the source of entangled states is turned on.
It is shown that the data recorded in similar experiments performed in 2012, repeated in 2018 and improved in 2019 are consistent with the proposed transient deviations hypothesis, but they are insufficient to validate or
refute it.
Round 2
Reviewer 1 Report
I have no further comment on the manuscript and/or on the Author's response. I had already said everything in my earlier report and there is nothing dramatic that happened in the meantime to withdraw my earlier decision.
Reviewer 2 Report
The authors have addressed my questions and they have clarified the paper. It is now clear that the AB model alone is not able to explain the results of the loophole-free experiments. An additional exploitation of experimental imperfections is needed. In contrast to what the authors write in section 2.2, I would find it interesting to consider what local realistic models would be able to explain the Bell violation as observed in the loophole-free experiments.
In my opinion, this paper is innovative and well written. It explains a model and it suggests an experiment to test the model. I think that many readers of Entropy will find it interesting to think about the questions addressed in this paper, namely whether the hypothesis of transient violation of Bell’s inequality can allow one to maintain a local realist position, and how this hypothesis can be tested. I do not agree with all statements in this manuscript. But I think that this manuscript should be published. I believe it is a valuable contribution to the discussion about Bell’s inequality.